# Immunotherapy and Targeted Therapy for Advanced Biliary Tract Cancer: Adding New Flavors to the Pizza

**DOI:** 10.3390/cancers15071970

**Published:** 2023-03-25

**Authors:** Marcello Moro Queiroz, Nildevande Firmino Lima, Tiago Biachi de Castria

**Affiliations:** 1Oncology Center, Hospital Sírio-Libanês, 115 Dona Adma Jafet Street, São Paulo 01308-050, SP, Brazil; 2Moffitt Cancer Center, 12902 USF Magnolia Drive, Tampa, FL 33612, USA; 3Morsani College of Medicine, University of South Florida, 12901 Bruce B. Downs Blvd., Tampa, FL 33612, USA

**Keywords:** biliary tract cancers, intrahepatic cholangiocarcinoma, extrahepatic cholangiocarcinoma, hilar cholangiocarcinoma, gallbladder cancer, immunotherapy, targeted therapy

## Abstract

**Simple Summary:**

Biliary tract cancers (BTCs) are a rare pathology. In the era of precision oncology, the development of next-generation sequencing (NGS) allowed a better understanding of molecular differences between each tumor. Alterations in genes such as the *FGFR*, *HER2*, *IDH1*, and *BRAF*, result in cancer development, growth, and proliferation. Recent drug development allowed for the use of medications that can target these alterations and inhibit cancer progression. Additionally, the understanding of how the immune system interacts with cancer cells also resulted in the use of immunotherapy in this difficult scenario, although we still do not know how to select the patients with a higher chance of response. Here, we review the most recent data regarding targeted treatment and immunotherapy in the scenario of BTC treatment, discussing not only the role that these medications have in modern oncology, but also the future perspectives for this challenging disease.

**Abstract:**

Biliary tract cancers (BTCs) are a rare pathology and can be divided into four major subgroups: intrahepatic cholangiocarcinoma, extrahepatic cholangiocarcinoma, hilar cholangiocarcinoma, and gallbladder cancer. In the era of precision oncology, the development of next-generation sequencing (NGS) allowed a better understanding of molecular differences between these subgroups. Thus, the development of drugs that can target these alterations and inhibit the abnormal pathway activation has changed the prognosis of BTC patients. Additionally, the development of immune checkpoint inhibitors and a better understanding of tumor immunogenicity led to the development of clinical trials with immunotherapy for this scenario. The development of biomarkers that can predict how the immune system acts against the tumor cells, and which patients benefit from this activation, are urgently needed. Here, we review the most recent data regarding targeted treatment and immunotherapy in the scenario of BTC treatment, while also discussing the future perspectives for this challenging disease.

## 1. Introduction

Biliary tract cancers (BTCs) are a rare pathology, with an estimated number of new cases and deaths of 12,130 and 4400, respectively [1], in 2022 in the United States. They are usually diagnosed in older patients, with a median age ranging from 70 to 72 years, and have a higher incidence in Southeast Asia, primarily because of their association with parasitic infections by Opisthorchis viverrini and Clonorchis sinensis. In close relation to all BTCs is cholangiocarcinoma (CC), a neoplasm derived from gland cells (adenocarcinoma), which can be divided into four diverse subgroups: intrahepatic cholangiocarcinoma (IHCC), extrahepatic cholangiocarcinoma (EHCC), hilar cholangiocarcinoma (Klatskin tumors), and gallbladder (GB) cancer. Other less common subtypes are sarcomas, lymphomas, and small cell cancers, which are not usually included in the statistics [1,2]. Although it can vary depending on the subtype, the prognosis of this disease remains dismal despite recent advances in treatment, with a five-year relative survival rate for IHCC of 24% and 2% for localized and distant disease, respectively, and 17% and 2% for EHCC [3].

Treatment of localized disease consists of surgery with curative intent, with or without adjuvant therapy with capecitabine, based on data from the BILCAP trial, which demonstrated benefits in recurrence-free survival (RFS), without benefit in overall survival (OS) in the intent-to-treat population [4]. The cure rate for patients with early-stage disease ranges from 60% to 70%, with factors such as lymph node positivity and R1 resection associated with decreased RFS and OS [5,6,7]. Nonetheless, approximately 60–70% of patients present with metastatic or unresectable disease at diagnosis, whose main treatment options consist of systemic treatment or palliative locoregional strategies when necessary [8]. Worldwide treatment guidelines define the combination of gemcitabine and cisplatin (GemCis) as the standard first-line treatment, based on data from the ABC-02 trial. This study showed OS benefit with the combination when compared to gemcitabine (11.7 m vs. 8.1 m, HR 0.64, *p* < 0.001), without the addition of substantial toxicity [9,10,11]. More recently, the addition of durvalumab to GemCis is considered the standard of care in the first-line treatment scenario, based on the data from the TOPAZ-1 trial [12,13,14]. Another triplet regimen was recently tested in the SWOG-1815 trial; however, it did not show the same benefit as the one with immunotherapy. In this study, the addition of nab-paclitaxel to GemCis did not achieve a statistically significant improvement in overall and progression-free survival over GemCis alone, in patients with newly diagnosed, advanced BTC [15,16].

Although the second line represents a difficult scenario, with a prior median OS (mOS) of 7.2 months and median progression-free survival (mPFS) of 3.2 months, significant advances have been made in recent years due to the tumor characterization and development of targeted therapies [17]. There are two main standard options of chemotherapy when no actionable mutations are identified: FOLFOX or liposomal irinotecan combined with fluorouracil (5FU) and leucovorin (LV). Treatment with FOLFOX is supported by the data published in ABC-06. This phase III randomized trial demonstrated improvement of OS with FOLFOX compared to active symptom control, in patients previously treated with GemCis (6.2 m vs. 5.3 m, HR 0.69, *p* = 0.031) [18]. The phase IIb NIFTY trial randomly assigned patients previously treated with GemCis to receive 5FU, LV, and liposomal irinotecan or 5FU/LV. The experimental group experienced an increase in median PFS (7.1 m vs. 1.4 m, HR 0.56, *P* = 0.0019), at the expense of an increase in serious adverse events (AE), especially neutropenia [19].

Nevertheless, with the diffusion of next-generation sequencing (NGS), tests and description of somatic alterations prevalent in biliary cancers, targeted therapy has become an option increasingly studied in the advanced scenario. Among patients with BTC, the most frequent somatic mutations occur in the genes TP53, CDKN2A/B, KRAS, and SMAD4; however, a lower prevalence (<5%) is described in targetable mutations such as IDH1/2, FGFR2, BRAF, PIK3CA, and NTRK [20,21,22,23,24,25,26]. As previously mentioned, there are differences in mutation profiling depending on the subgroups of the BTC analyzed. For example, mutations in BAP1, CDK2NA, ARD1A, FGFR1–3, and MET are more frequent in IHCC, while KRAS is mostly described in EHCC and TP53, PIK3CA, HER2, BRAF, EGFR in GB. Additionally, the presence of mutations in IDH1/2 is most likely exclusive to IHCC and the absence of mutations in FGFR1–3, MET and EGFR are also most likely exclusive to EHCC [27].

In this article, we reviewed the recent therapy advances for the treatment of BTC (Table 1), describing their mechanism of action and discussing the main clinical trials published to this date. 

## 2. Fibroblast Growth Factor Receptors (FGFRs)

Fibroblast growth factor receptors (FGFRs) are a family of four transmembrane receptors with intracellular tyrosine kinase domains (FGFR5 acts as a co-receptor of FGFR1, lacking an intracellular domain receptor), mediated by growth factors of fibroblasts (FGFs), constituting an important signaling pathway in organ systems through the regulation of cell proliferation, angiogenesis, migration, and DNA repair (Figure 1). 

Molecular profiling of 4853 solid tumors by NGS revealed that FGFR alterations were found in 7.1% of cancers, with more than half being gene amplification (66%), followed by mutations (26%) and rearrangements (8%). The most common cancers to harbor any FGFR alterations were urothelial (32%), breast (18%), endometrial (13%), squamous lung cancers (13%), and ovarian cancer (9%), suggesting that targeting FGFR may be an interesting therapeutic strategy across multiple tumor subtypes [28]. In CC, there is a predominance of fusions (3.5%), mainly involving FGFR2, present in up to 16% of intrahepatic tumors. Amplifications and activating mutations are present in 2.6% and 0.9% of bile duct cancer, respectively [28,29]. A retrospective study evaluated clinical and molecular features of FGFR-altered CCs and identified a predominance in women, with a mean age of 57 years, and up to 95.6% of intrahepatic lesions. Most of the lesions were in early stages (I or II), with FGFR2 fusion being the most common alteration (68%), followed by FGFR2 mutations (22%). The most frequently described fusion partner to FGFR2 was BICC1 (28.3%), correlating to improved ORR when compared to non-BICC1 fusion (42.9% vs. 30.8%) when treated with FGFR-selective inhibitors [30]. In addition to this trial, others also suggest a more indolent evolution among tumors that harbor FGFR alterations [31]. Based on the significant prevalence of this molecular alteration and the possibility of targeting it, several therapeutic trials have been developed for the treatment of BTC with FGFR alterations, in the last years.

The phase II FIGHT-202 trial evaluated the use of pemigatinib in the second line for patients with CC harboring FGFR2 fusions or rearrangements and demonstrated an ORR of 35.5% and an impressive mOS of 21.1 months, thus emerging as an option for patients with tumors that progressed to a platinum-based regimen [32]. Considering these promising results, the FIGHT-302, a phase III study, will soon bring the results of the comparison between pemigatinib compared to the standard first-line regimen of GemCis in patients with advanced disease [33]. Another FGFR inhibitor, infigratinib, was studied in a phase II trial that included only patients with FGFR2 fusion or rearrangement IHCC and showed an mOS of 12.2 m and ORR of 23%, obtaining accelerated approval for this scenario by the Food and Drugs Administration (FDA) in May 2021 [34,35]. Erdafitinib, a pan-FGFR inhibitor approved in 2019 by the FDA for the treatment of locally advanced or metastatic FGFR3 or FGFR2 urothelial carcinoma, was also recently studied in CC. Feng et al. presented the updated analysis of the LUC2001 trial at ASCO 2022, a phase IIa study that screened 232 patients with previously treated advanced CC. Thirty-nine patients harbored FGFR alterations (16.8%) and 22 (9.5%) were eligible for treatment with erdafitinib. After a median follow-up of 22.4 months, the ORR was 40.9%, with an mOS and mPFS of 40.2 and 5.6 months, respectively [36].

Deranzantinib and futibatinib are new FGFR inhibitors that were recently studied in patients with CC harboring FGFR alterations. Data of the phase II FIDES-01 trial was presented at the 2022 ASCO Gastrointestinal Cancers Symposium, describing the treatment of 28 patients with derazantinib, a potent anti-FGFR 1–3 oral kinase inhibitor. Patients with FGFR-mutated or -amplified IHCC who progressed after at least one line of standard chemotherapy presented with a 75.7% DCR, with a mPFS and mOS of 8 and 16 months, respectively. The authors also described a clinically meaningful anti-tumor efficacy observed across all types of genetic aberrations [37]. Futibatinib, a highly selective and irreversible FGFR 1–4 inhibitor, also showed meaningful results, with a DCR of 78.6% and exactly double ORR when compared to deranzantinib, (42% vs. 21%). Moreover, recent updates also describe a mPFS and mOS of 9 and 21.7 months, respectively [38].

## 3. Human Epidermal Growth Receptor 2 (HER2)/ERB-B2 Receptor Tyrosine Kinase 2 (*ERBB2*)

Human epidermal growth receptor 2 (HER2) is a transmembrane glycoprotein regulated by the *ERBB2* gene (located on chromosome 17), and it is fundamental in regulating cell proliferation. When amplified, the gene causes overexpression of HER2, triggering tumor development and configuring a worse prognosis, with a higher risk of recurrence, and worse survival [39]. Different HER2 targeting agents were established as effective standard therapy for breast and gastric cancer, adding a better efficiency, improved outcomes with overall survival, and a good safety profile [40].

Present in about 30% and 22% of breast and gastroesophageal cancer cases, respectively, HER2 amplification has been proposed as an oncogenic driver, allowing for the inclusion of new drugs in the treatment of other tumors, such as colon, non-small cell lung, urothelial, and BTC [41,42]. Overexpression of HER2 by IHC is found in about 57% of bile duct tumors, with 5%, 16%, and 12% in EHCC, GB, and ampulla of Vater, respectively, being under-expressed or even absent in IHCC. Moreover, HER2 amplification by FISH is present in 5.6% of cases, 67% with 2+ HER2, and 100% with +3 [43]. Mondaca et al. described 517 patients with BTC that were evaluated with NGS and reported that alterations in *ERBB2* were identified in 5.4% of cases, with a description of HER2 amplification in 2.7% of the cases, 2.3% of *ERBB2* mutation, and 0.4% of concurrent amplification and mutation. Additionally, the prevalence of *ERBB2* alterations was significantly higher in GB (12.6%) when compared to EHCC (7.5%) and IHCC (2.2%). Patients with ERBB2 alterations also presented co-alterations, such as microsatellite instability, high (MSI-H) in 7% of the cases, TP53 mutations in 54%, and PIK3CA in 21% of the cases, all significantly higher when compared to patients without *ERBB2* alterations. KRAS amplification/mutation, a potential resistance mechanism of HER2-targeted therapy, had a lower prevalence in patients harboring *ERBB2* alteration (7% vs. 16%, *p* = 0.2866), which highlights the relevance of this pathway in BTC [44].

The HER2 overexpression has been configured as a BTC subtype, representing a neoplasm with a bigger capacity for invasiveness, proliferation, metastasis, low response to chemotherapy, and correlating with a worse prognosis [45]. HER2 overexpression results in several downstream signaling pathways, for example, the phosphatidyl inositol 3 kinase (PI3K)/protein kinase B (AKT) signaling pathway, the most important in HER2 heterodimerization. This upregulation is also associated with worse outcomes [46]. The AKT, expressed in approximately 85% of EHCC, is activated by the mammalian target of rapamycin (mTOR) that phosphorylates. The homologous tensin phosphatase (PTEN) is a gene that suppresses the PI3K/AKT pathway, decreasing the levels of cyclin D1, interfering with the G1 phase of the cell cycle, and greatly impacting survival. This pathway has been the target of several therapies under development [47,48].

The SUMMIT trial, a phase II “basket” study of multi-histology tumors harboring somatic *HER2* alterations, evaluated 25 patients with BTC treated with neratinib, an irreversible pan-HER2 tyrosine kinase. The authors reported improvements in mPFS (2.8 months), mOS (5.4 months), and ORR (12%) [49]. MyPathway trial, another basket trial, included 39 patients with previously treated BTC, with *HER2* amplification/overexpression treated with trastuzumab and pertuzumab. Treatment was well tolerated and resulted in an ORR of 23%, PFS of 4 months, and mOS of 10.9 months [50].

In the recent years, a new therapeutic strategy targeting HER2 alterations, known as an antibody–drug conjugate (ADC), is gaining space due to promising results. Trastuzumab Deruxtecan (T-DXd), a HER2-directed antibody and topoisomerase inhibitor conjugate, showed impressive responses in patients with previous BTC and HER2-positive (IHC 3+, IHC 2+/ISH+) and HER2-low (IHC/ISH of 0/+, 1+/+, 2+/−) cancers, during the first data presentation at the ASCO 2022. The ORR in HER2+ patients was 36.4% with DCR, mPFS, and mOS of 81.8%, 4.4 months, and 7.1 months, respectively. Encouraging results were also described for patients described as HER2-low, with an ORR, DCR, mPFS, and mOS of 12.5%, 75.0%, 4.2 months, and 8.9 months, respectively. However, it is important to highlight that this drug was associated with a high incidence of grade 3–4 events (81.3%), with 25% evolving into interstitial lung disease (ILD), a critically known risk of T-DXd, that resulted in two deaths in this cohort [51].

Still, in HER2 overexpression and advanced disease scenario, Zanidatamab (ZW25), a novel bispecific anti-HER2 target, was tested in an isolated regimen in second-line treatment. An interesting ORR of 47% was identified, in addition to a good safety profile, without any grade 3 AEs [52]. Further, encouraging results are expected from HERIZON-BTC-01 (NCT04466891), a phase 2b trial that will better evaluate the efficacy of this drug in patients that received at least one prior gemcitabine-containing systemic chemotherapy. Additionally, the ZW25-201 trial (NCT03929666), a phase 2b basket trial, will evaluate the role of Zanidatamab plus the investigator’s choice of chemotherapy in gastrointestinal tumors, currently in the first-line setting.

## 4. Isocitrate Dehydrogenase 1 (IDH1) 

IDH1 and IDH2 are key proteins in cellular metabolism, redox states, epigenetic regulation, and DNA repair [53]. Mutant IDH proteins result in an abnormal enzymatic activity, allowing them to convert α-ketoglutarate (αKG) to 2-hydroxyglutarate (2HG), which inhibits the activity of multiple αKG-dependent dioxygenases. This inhibition results in alterations in cell differentiation, survival, extracellular matrix maturation, and has a fundamental role in the inhibition of the homologous recombination repair mechanism (Figure 2). Pre-clinical data demonstrated that the IDH1 mutations (mIDH1) act by blocking the liver progenitor cells from undergoing hepatocyte differentiation through the production of 2HG and suppression of HNF-4α, a regulator of hepatocyte identity and quiescence, thus affecting the liver progenitor cell differentiation and proliferation [54] (Figure 2).

mIDH1 is relatively frequent in patients with IHCC. A recent systematic review that included 45 publications that reported the frequency of mIDH1 among a total sample of 5393 patients with CC, showed a prevalence of 13.1% in IHCC compared to 0.8% in EHCC (*p* < 0.0001). Considering patients with IHCC, the prevalence in Asian centers was statistically lower (8.8%) compared to non-Asian centers (16.5%), particularly in the United States (18%). However, this review lacks statistical power to correlate the frequency of this mutation with gender and age. Additionally, the most common co-mutations presented with mIDH1 IHCC were AT-rich interactive domain-containing protein 1A (ARID1A), BRCA1-associated protein 1 (BAP1) (loss or mutation), and polybromo1 (PBRM1) [55].

Ivosidenib, a potent small-molecule inhibitor of mIDH1 tumors, was first approved for treating patients with advanced acute myeloid leukemia [56]. After a positive phase I dose escalation and expansion study of this medication in patients with previously treated IDH1-mutant advanced CC, this drug was further tested in a phase III randomized trial (ClarIDHy) [57,58]. Abou-Alfa et al. described the treatment of 185 patients with IDH1-mutant cholangiocarcinoma, who had progressed in previous therapy and had up to two previous treatment regimens for advanced disease. Patients were randomized to ivosidenib or placebo, and the experimental arm showed an improvement in PFS (median 2.7 m vs. 1.4 m, HR 0.37, one-sided *p* < 0.0001), with a tolerable safety profile. So far, it was the first and only phase III randomized multicenter trial to demonstrate the clinical benefit of targeting IDH1 in this scenario. A possible strategy to optimize the effectiveness of IDH1 inhibitors is the association with other drugs, such as immunotherapy. Given this, olutasidenib, an IDH1 inhibitor, has been evaluated in solid tumors with IDH1 mutation, associated or not associated with other drugs, such as nivolumab (NCT03684811). A phase II study cohort has been evaluating olaparib use, a PARP inhibitor, in patients with CC, with mutated IDH1 or IDH2 (NCT03212274). 

## 5. BRAF Proto-Oncogene (BRAF)

The *BRAF* gene encodes a cytoplasmic serine/threonine kinase with a key role in regulating the mitogen-activated protein kinase signal transduction pathway [59]. The prevalence of *BRAF* gene mutations in BTC ranges between 5 and 7%, with a higher prevalence when considering only IHCC and one of the most common mutations in this gene is the BRAF V600E [60,61,62]. This missense mutation, which results in the substitution of the amino acid valine to glutamic acid in codon 600, is a class I *BRAF* mutation and exhibits a robust kinase activity by stimulating monomeric activation of *BRAF*, resulting in MAPK pathway activation [63]. Considering IHCC and the presence of *BRAF* V600E mutation, this alteration is associated with a higher tumor stage at the time of resection, higher probability of lymph node involvement, and worse long-term OS [64].

Tumors harboring this mutation were first treated with single-agent BRAF inhibitors, however, the questionable safety profile and the absence of durable responses resulted in trials combining this drug class with MEK inhibitors [65,66]. The combination of BRAF and MEK inhibitors resulted in increased PFS, and OS compared to a single BRAF inhibitor, consolidating the combination as the standard of care in patients harboring the mutation BRAF V600E [65]. 

Regarding BTC, the Rare Oncology Agnostic Research (ROAR) basket trial was the first prospective analysis of a cohort of patients with BRAFV600E-mutated BTC, treated with BRAF and MEK inhibitors, dabrafenib and trametinib, respectively [67]. After including 43 patients with unresectable, metastatic, or locally advanced BTC treated with at least one previous line, the independent reviewer-assessed ORR was 47%, with a manageable safety profile. In addition, the use of these medications was recently approved by the FDA for any unresectable or metastatic solid tumors with BRAF V600E mutation [68].

## 6. Immune Checkpoint Inhibitors (ICIs)

In the recent years, the development of monoclonal antibodies that act against immune checkpoints shifted the paradigm in the management of solid tumors. The main examples are antibodies that target the membrane protein programmed cell death protein 1 (PD-1) and its ligand (PD-L1), of which the pathway is related to evasion and tolerance mechanisms that are involved with activated T cells with cytotoxic capabilities [69,70]. Moreover, the combination of immunotherapy with chemotherapy may represent a way to overcome the difficult scenario of advanced BTC treatment. Some studies have suggested that chemotherapy may upregulate checkpoint expression and change immune cell infiltrate [71,72,73]. Not only does chemotherapy promote tumor immunity by inducing immunogenic cell death, but it also acts by disrupting strategies that tumors use to evade immune recognition [74].

Regarding BTC, the first interim analysis of the TOPAZ-1 trial was presented at the 2022 American Society of Clinical Oncology (ASCO) Gastrointestinal Symposium, and later published in the New England Journal of Medicine Evidence, describing the use of durvalumab (anti-PD-L1) combined with chemotherapy, versus the standard first-line treatment with placebo. After the inclusion of 685 patients with untreated metastatic BTC, the addition of the ICI to GemCis resulted in increased mOS (12.9 m vs. 11.3 m, HR 0.76) in the most recent update, PFS (HR 0.75, *p* = 0.001), ORR (26.7% vs. 18.7%) and the duration of response when compared to chemotherapy alone, with a manageable safety profile [12,13,14].

On the second line of treatment, pembrolizumab (anti-PD-1) is an option for previously treated patients with microsatellite instability (MSI-H) or deficient mismatch repair proteins (d-MMR). The use of this ICI is based on data from KEYNOTE-158, a study that demonstrated the benefit of this drug in patients that harbor MSI-H or d-MMR, irrespective of solid tumor primary site, resulting in the first tumor-agnostic FDA drug approval granted on May 23, 2017 [75,76]. However, these alterations are described as infrequent in BTC, with the frequency of MSI-H being 5% each for GB and EHCC, and 10% each for IHCC and ampullary carcinoma [77].

More recently, Kim et al. published data from 54 patients with BTC, who previously progressed to at least one line and were treated with nivolumab [78]. The central independent review found an ORR of 11% and a DCR of 50%, with all patients who responded to treatment harboring pMMR. The intention-to-treat population presented with an mPFS of 3.68 months and an mOS of 14.28 months. Biomarker analysis was made, and PDL1 expression in tumors was significantly associated with prolonged PFS, with cut-offs of ≥1% and >10%, but not with OS. The potential prognostic value of PD1 expression in tumor-infiltrated lymphocytes (TILS) was also evaluated, but no correlation with clinical outcomes was achieved, even when this biomarker was combined with PD-L1 expression on the tumor.

## 7. ICI Biomarkers

Using efficient biomarkers that may have a prognostic role in predicting resistance or susceptibility to immunotherapy is a valuable tool for selecting who may benefit from specific neoadjuvant or adjuvant treatments. A study sought to determine immune response patterns in BTC samples through DNA methylation, RNA expression, and immunohistochemistry analysis. Samples enriched with T lymphocytes were correlated with the activation of inflammatory and immune checkpoint pathways, conferring better survival and greater vulnerability to ICIs. In contrast, a tumor microenvironment with M2 macrophage infiltrates reduced MHC class I and the loss of β2-microglobulin, and the absence of CD8 T cells leads to an immunologically silent tumor [79].

Tumor mutational burden (TMB) and MSI represent essential biomarkers for ICI response, predicting possible resistance to treatment in several tumors. A Chinese group evaluated 432 samples of gastrointestinal tumors (four of which were gallbladder and biliary tract cancer) and identified a low rate of chromosomal instability (18%), as well as a low TMB, in line with previous data, showing that less than 3% of BTCs have more than 15 mutations/Mb [80,81]. 

In the analysis by NGS, there are alterations in DNA repair genes (ATM, ATR, BRCA1, BRCA2, FANCA, MLH1, MSH2, PALB2, and POLE) and caretaker genes (BAP1 and TP53) in approximately 14% and 63% of patients with GBC, respectively. Both presented high TMB (≥19.5 mutations/Mb) and were PD-L1-positive (16%), although predominantly low [82]. In particular, somatic and germline mutations in the DNA polymerase (POL) gene, specifically in exonuclease domains (EDM) of POLE and POLD1, are more prevalent in endometrial and colon tumors. These are related to hypermutated phenotype, with high TMB and upregulation of immune checkpoint inhibitors [83]. However, some analyses have identified cases of mutation in the EDM of POLE with ultra-mutated and MSS phenotypes [84].

PD-L1 expression has been reported in 9.1–72.2% of patients with BTC, a membrane protein directly correlated with the way in which immunotherapy works, enhancing the immune response against the cancer cell [85,86,87]. Additionally, although the use of immunotherapy in the advanced scenario is approved regardless of PD1/PD-L1 expression, higher PD-L1 expression is associated with therapeutic response to immunotherapy, even in BTC [73,88,89]. However, even though there is a rationale to use PD-L1 as a biomarker of response, there is a long way to go for this to be achieved. Alternately, there are data on lung cancer showing that low-*PD-L1* copy-number tumors display reduced *PD-L1* expression, reduced PD-L1 tumor cell staining, and an immunologic cold tumor microenvironment, and could thus possibly lead to less response to immunotherapy [90].

## 8. Epidermal Growth Factor Receptor (EGFR)

EGFR is an essential transmembrane tyrosine kinase involved in activating the RAS/RAF/MAPK and AKT/mTOR signal transduction pathways, which are directly related to both cell proliferation and cell death [91]. Inhibition of EGFR has been a widely studied and actionable target for several types of tumors, including non-small cell lung and colon cancer, using either tyrosine kinase inhibitors (TKIs), such as osimertinib and erlotinib, or antibodies, such as cetuximab and panitumumab [92,93].

The expression of EGFR can vary depending on the analysis methodology. Through immunohistochemistry analysis, Shafizadeh et al. identified or strong (+2/+3) EGFR expression in approximately 59% of BTC samples, with a prevalence of 68%, 58%, and 33% in EHC, IHC and GB, respectively. When analyzed by FISH, 46% of the samples showed increased EGFR gene copy numbers. The same author identified worse 5-year survival outcomes among those expressing EGFR (20% vs. 60%) [94].

Given the high prevalence and the prognostic impact of EGFR in BTC, several studies have evaluated the efficacy of targeting EGFR, using either tyrosine kinase inhibitors (EGFR-TKIs) or monoclonal antibodies directed to EGFR (EGFR-mAbs). Although these targeted drugs have shown promising results in several types of tumors (NSCLC, colon, head, and neck), applying these treatments did not reproduce encouraging results in BTC [95]. The BINGO trial, a randomized phase II trial, evaluated the association between cetuximab (EGFR-mAbs) with gemcitabine, and oxaliplatin in patients with locally advanced (non-resectable) or metastatic BTC. The addition of the TKI did not result in significant gains in progression-free survival (6.1 vs. 5.5 months), overall survival (11 vs. 12.4 months), or objective response rate (24% vs. 23%) compared to the chemotherapy-alone group. Furthermore, serious adverse events were reported in the antibody group [96]. Similar results were found in the PICCA trial, which evaluated the gemcitabine and cisplatin plus panitumumab regimen when stratified by KRAS wild type [97]. When evaluating the association with the TKI erlotinib, the experimental group, although presenting a significant gain in objective response rate (30% vs. 16%, *p* = 0.005), did not show a significant gain in progression-free survival (5.8 vs. 4.2 months, *p* = 0.087) or overall survival (9.5 vs. 9.5 months, *p* = 0.611) [98]. 

Given the lack of statistically significant benefits in objective response rate, progression-free survival, and overall survival, a targeted therapy directed at EGFR (whether TKI or antibodies) has not been included in the therapeutic arsenal of BTC with overexpressed EGFR.

## 9. Chimeric Antigen Receptor T Cell (CAR-T Cell)

Chimeric antigen receptors (CARs) are an innovative type of immunotherapy with a well-established role in treating hematologic malignancies and have been extensively studied in solid tumors. This treatment involves collecting T cells from the patient, which are then genetically engineered to express chimeric antigen receptors (CAR-T cells) capable of identifying specific proteins on the surface of tumor cells. Once infused into the patient, these modified cells can target and destroy the cancer cells [99].

Various receptors found on the surface of biliary tract cancer (BTC) cells have been targeted using CAR-T-cell therapy. For instance, in a phase I trial, EGFR-specific CAR-T-cell therapy was evaluated in 19 patients with unresectable or metastatic BTC refractory to chemotherapy. Following infusion of a conditioning treatment with nab-paclitaxel and cyclophosphamide, 10 patients achieved stable disease and one complete response, leading to a 4-month progression-free survival. Among the primary side effects were fever, lymphopenia, and thrombocytopenia, attributed to the conditioning treatment [100].

In a phase I study by Feng et al., CART–HER2 immunotherapy was evaluated in 11 patients with BTC or advanced/metastatic pancreatic cancer with HER2 expression. Encouraging results were obtained, with a median progression-free survival of 4.8 months [101]. Other targets have also been explored using CAR-T-cell therapy, such as Mesothelin, CD133, Claudin 18.2, and MUC-1, with promising results in patients with BTC and other solid tumors [102,103,104,105].

## 10. Discussion

Precision oncology advancements allowed for the development of immune and targeted therapy, shifting the cancer treatment paradigm in the last years. Understanding the mechanism of oncogenesis by a driver mutation, when possible, or the characteristics that influence the tumor microenvironment and its response to immunotherapy, changed the treatment of metastatic disease from a more populational point of view to a personalized one for each patient and tumor. The main treatment options for the treatment of BTC are described in Table 1.

The possibility of using NGS to understand characteristics such as tumor mutational burden (TMB), microsatellite status, and pathological mutations, raised the question of when these tests should be done, as most of them are accompanied by a high financial burden. In addition to the high cost of this test, in many countries, NGS is not covered by insurance, creating an important barrier to the disponibility of this technology. Consequently, only a portion of the population that can pay for the test will benefit from it. Considering BTC, not all mutations have a targeted therapy, and the treatment options for the ones that can be targeted are restricted to the second-line treatment of metastatic disease. It is reasonable to think that at some point, these medications will be tested in previous scenarios, including for localized disease, as they showed impressive responses when compared to standard treatment, making the NGS essential for management choice in advanced scenarios. The most frequent genetic aberrations in targetable pathways of interest in BTC are shown in Table 2. Nevertheless, although data regarding biomarkers of response from the TOPAZ-1 are not available, there is an understanding that high TMB could correlate to a better response to ICI in some tumor types, thus justifying doing these tests at the diagnosis of the metastatic disease [106]. However, there is a lack of information regarding reliable biomarkers in response to ICI in BTC, showing the importance of including an analysis of biomarkers that could correlate to better outcomes.

Knowing the molecular profile of BTC guides personalized treatment and gives information on prognosis. A study published by Javle et al. showed that patients with IHCC, harboring mutations in TP53 and KRAS, had a significantly inferior OS. This same study also demonstrated that patients whose tumors had FGFR genomic alterations had superior OS with FGFR-targeted therapy versus standard regimens, and that targeted therapy in IHCC was associated with a numerically higher OS (*p* = 0.07) [24].

Compared to chemotherapy, targeted therapy was consistently associated with a better ORR and disease control rate (DCR) in the second line or later, with more than double DCR in patients with BRAF mutations treated with BRAF and MEK inhibitors, for example [67]. Additionally, therapies targeting HER2 alterations are showing impressive responses. However, the new medications, known as ADCs, are reformulating the way that we analyze HER2-positive tumors, as even HER2-low patients can present with considerable responses. However, despite the initial encouraging results of this new medication, the complicated safety profile, with a high incidence of complications such as ILD leading to death, may present a challenge.

In the first-line setting, although the absolute gain of OS with the addition of durvalumab to conventional chemotherapy was 1.3 months, this difference was statistically significant. It is important to note that the experimental arm was compared to the standard scheme of cisplatin plus gemcitabine, a well-known active scheme in BTC, with the MSI status missing for approximately half of the population and the curves showing a more discrepant difference after 6 months of maintenance therapy. Regarding immunotherapy in the second line, we still do not have a randomized phase III study compared to chemotherapy, and pembrolizumab is approved only for a small population of patients with MSI or dMMR tumors, based on an agnostic approval from a pan-tumor study. Additionally, although the nivolumab study described biomarker analysis in patients that responded, we do not have a clear and reliable biomarker that could predict ICI benefit in this population.

## 11. Conclusions

In conclusion, despite the protracted development in the treatment of BTC in the past years, the development of NGS and molecular biology enables a better understanding of this disease. As a result, targeted treatment and immunotherapy are gradually changing the paradigm of patients with metastatic BTC, despite most trials being phase II. Further phase III biomarker-based studies in the first line are needed to select better patients most likely to benefit from these strategies. Soon, the answer to these questions may benefit patients in the metastatic scenario and those with locally advanced disease, thus providing a better chance of cure for this challenging disease.

## Figures and Tables

**Figure 1 cancers-15-01970-f001:**
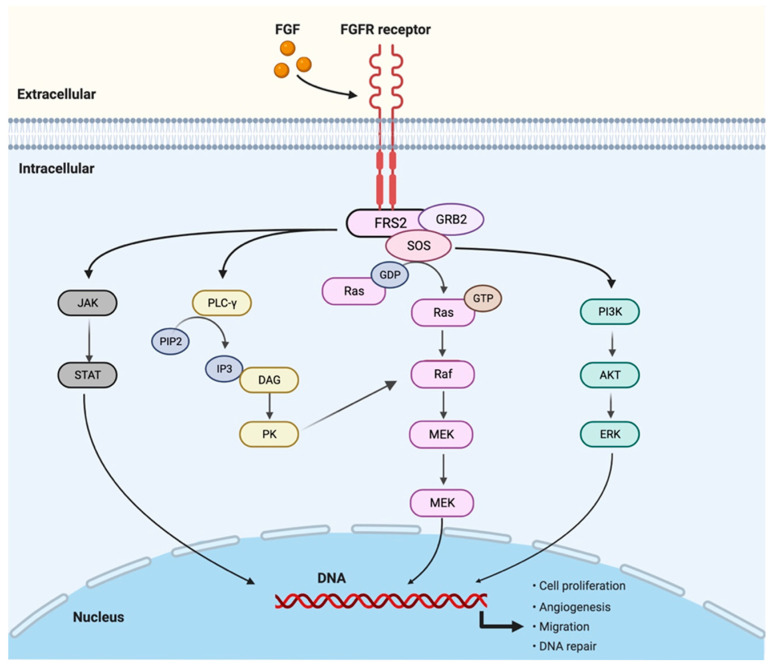
FGFR pathway: The FGFR dimerizes upon ligand (FGF) binding, activating the downstream cascade of signaling pathways. The main pathways activated are the phosphorylation of the signal transducer and activator of transcription (STAT), the PLCγ activation of the DAG–PKC and IP3–Ca^2+^ cascade, resulting in DNA transcription, the mitogen-activated protein kinase (MAPK) pathway, and the phosphoinositide-3-kinase (PI3K/Akt) pathway. Figure created with BioRender.com.

**Figure 2 cancers-15-01970-f002:**
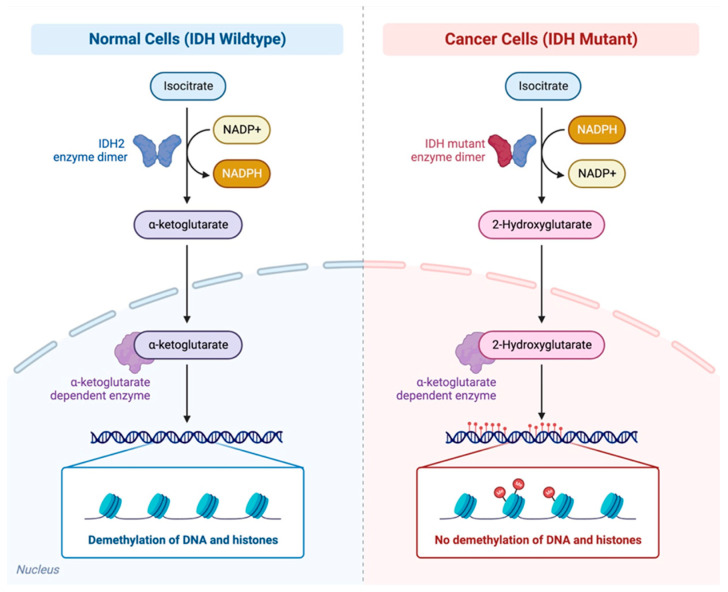
Differences between normal cells with IDH wildtype and cancer cells with IDH mutant. Mutant IDH proteins result in an abnormal enzymatic activity, allowing them to convert α-ketoglutarate (αKG) to 2-hydroxyglutarate (2HG), which inhibits the activity of multiple αKG-dependent dioxygenases. These alterations result in the impairment of the DNA, and histone demethylation. Figure created with BioRender.com.

**Table 1 cancers-15-01970-t001:** Studies that included advanced biliary tract cancer patients and their main characteristics. 5-FU—5-fluoracil; AE—adverse events; Amp—ampulla; ASC—active symptom control; Cis—cisplatin; CC—cholangiocarcinoma; DCR—disease control rate; dMMR—deficient mismatch repair; EH—extra-hepatic; FGFR2—fibroblast growth factor receptors; FOLFOX—5-fluoracil, leucovorin and oxaliplatin; GB—gallbladder; Gem—gemcitabine; HER2—human epidermal growth receptor 2; IH—intra-hepatic; LV—leucovorin; m—months; mPFS—median progression-free survival; mOS—median overall survival; ORR—overall response rate; T-DXd—trastuzumab deruxtecan.

Study	Treatment	Line	Design	Sample	Population	OS (m)	PFS (m)	ORR	Grade 3–4 AE
TOPAZ-1	CisGem + Durvalumab vs. CisGem	First	Phase III	685	IH 55.7% EH 19.4% GB 24.9%	12.8 vs. 11.5	7.2 vs. 5.7	26.7% (DCR 85.3%)	75.7%
ABC-02	CisGem vs. Gem	First	Phase III	410	BD 57.8% GB 36.9% Amp 5.3%	11.7 vs. 8.3	8.4 vs. 6.5	26% (DCR 81.4%)	68.8%
ABC-06	FOLFOX vs. ASC	Second	Phase III	162	IH 47% EH 23% GB 21% Amp 9%	6.2 vs. 5.3	4	5% (DCR 33%)	69%
NIFTY	Liposomal irinotecan + 5FU + LV vs. 5FU + LV	Second	Phase 2b	174	IH 39.8% EH 25% GB 35.2%	8.6	7.1 vs. 1.4	14.8% (DCR 64.8%)	42%
HERB	Trastuzumab deruxtecan (T-DXd)	Second	Phase II	32 (HER2+ 24, HER2 low 8)	IHCC 9.3% EHCC 18.7% GB 34.3% Amp 9%	HER2+ 7.1 HER2 low 8.9	HER2+ 4.4 HER2 low 4.2	HER2+ 36.4% (DCR 81.8%) HER2 low 12.5% (DCR 75%)	81.3%
SUMMIT	Neratinib	Second+	Phase II	25	IHCC 24% EHCC 20% GB 40% Amp 16%	5.4	2.8	12% (DCR 20%)	n.a
MyPathway	Trastuzumab + Pertuzumab	Second+	Phase 2b	39	IHCC 18% EHCC 18% GB 41% Amp 13% Undesignated 10%	10.9	4	23% (DCR 51%)	46%
ROAR	Dabrafenib + Trametinib	No other standard treatment options available	Phase II	43	IH 91% PH 2% GB 2% UN 2% Missing 2%	14	9	47% (DCR 82%)	40%
FIGHT-202	Pemigatinib	Second+	Phase II	146 (FGFR2 fusions or Rearrangements)	IH 98% EH 1% Other/missing 1%	21.1	6.9	35.5% (DCR 82.2%)	64%
NCT02150967	Infigratinib	Second+	Phase II	108	IH 100%	12.2	7.3	23% (DCR 84.3%)	64%
LUC2001	erdafitinib	Second+	Phase IIa	22	CC 100%	40.2	5.6	40.9%	68.2%
KEYNOTE-158	Pembrolizumab	Second+	Phase II	233	CC 9.4%	24.3	4.2	40.9%	14.6% (All population)
NCT02829918	Nivolumab	Second+ but no more than three previous lines	Phase II	54	IH 59% EH 9% GB 31%	14.24	3.8	11% (DCR 50%) All dMMR	17%
ClarIDHy	Ivosidenib vs. placebo	Third	Phase III	185	IH 90% EH 1% PH 3% UN 6%	10.8 vs. 9.7	2.7 vs. 1.4	2% (DCR 53.2%)	30%

**Table 2 cancers-15-01970-t002:** The frequency of the most frequent genetic aberrations in targetable pathways of interest in BTC. The frequencies were based on the publication of Valle J.W. et al. (Cancer Discovery, 2017) [107]. Abbreviations: GB—gallbladder; IHCC—intrahepatic cholangiocarcinoma; EHCC—extrahepatic cholangiocarcinoma.

Gallbladder	Extrahepatic Cholangiocarcinoma	Intrahepatic Cholangiocarcinoma
**TP53 mutation (47.1–59%)**	TP53 mutation (40%)	FGFR1-3 fusion, mutations and amplifications (11–45%)
**ERBB2/3 amplification (9.8–19%)**	KRAS mutation (8.3–42%)	TP53 mutation (2.5–44.4%)
**CDKN2A/B loss (5.9–19%)**	SMAD4 mutation (21%)	ARID1A mutation (6.9–36%)
**ARID1A mutation (13%)**	CDKN2A/B loss (17%)	MCL1 amplifications (21%)
**PIK3CA mutation (5.9–12.5%)**	ERBB2/3 amplification (11–17%)	IDH1/2 mutation (4.9–36%)
**KRAS mutation (4–13%)**	ARID1A mutation (12%)	CDKN2A/B loss (5.6–25.9%)
**NRAS mutation (6.3%)**	IDH1/2 mutation (0–7.4%)	KRAS mutation (8.6–24.2%)
**BRAF mutation (1–5.9%)**	PIK3CA mutation (7%)	SMAD4 mutation (3.9–16.7%)

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
