# Peer review of "Immunotherapy and Targeted Therapy for Advanced Biliary Tract Cancer: Adding New Flavors to the Pizza"

_cancers, 2023, doi:10.3390/cancers15071970_

Round 1

Reviewer 1 Report

The authors have set out to review the literature for targetted treatments for BTC. They have done a great job of summarizing the literature and should be commended. While there have been other publications on this in the past, this would be the most uptodate summary at the moment.

I only have some minor comments about the manuscript.

Minor comments:

1. Its important to reference your statistics in the introduction. Where are you getting the case number in the US for example.

2. It would be important to mention the use of immunotherapy now in BTC (TOPAZ-1 study) in the introduction as well (as it is now listed as a preferred regimen for BTC on the NCCN guidelines).

3. It would also be important to highlight the Gemcitabine/Cisplatin/nab-paclitaxel regimen (SWOG S1815)

4. Figure 4 is hard to read if you want to compare GB, EHCC and IHCC. I am wondering if a table would be better as it would allow for direct comparisons?

5. I think it would be worthwhile mentioning that cost of NGS sequencing could be a barrier to all of this. In many jurisdictions/countries, NGS sequencing is not covered so it would prevent access to many of these treatments. It could be good to use this as a platform for encouragement to get NGS sequencing funded in your country.

Author Response

Comment 1. “1. Its important to reference your statistics in the introduction. Where are you getting the case number in the US for example.”

Response 1. We would like to thank Reviewer #1 for his comments and suggestions. The reference was included as requested.

Comment 2. “2. It would be important to mention the use of immunotherapy now in BTC (TOPAZ-1 study) in the introduction as well (as it is now listed as a preferred regimen for BTC on the NCCN guidelines).”

Response 2. As recommended by the reviewer, we have briefly discussed the use of immunotherapy in the introduction:

          “More recently, the addition of durvalumab to GemCis is considered the standard of care in the first-line treatment scenario based on the data from the TOPAZ-1 trial.”

Comment 3. “3. It would also be important to highlight the Gemcitabine/Cisplatin/nab-paclitaxel regimen (SWOG S1815).”

Response 3. We further discussed the SWOG S1815 in the introduction:

“Another triplet regimen was recently tested in the SWOG-1815 trial; however, it did not show the same benefit as the one with immunotherapy. In this study, the addition of nab-paclitaxel to GemCis did not achieve a statistically significant improvement in overall and progression-free survival over GemCis alone in patients with newly diagnosed, advanced BTC.”

Comment 4. “4. Figure 4 is hard to read if you want to compare GB, EHCC and IHCC. I am wondering if a table would be better as it would allow for direct comparisons?”

Response 4. We changed the figure 4 to a table. See table 2.

Comment 5. “5. I think it would be worthwhile mentioning that cost of NGS sequencing could be a barrier to all of this. In many jurisdictions/countries, NGS sequencing is not covered so it would prevent access to many of these treatments. It could be good to use this as a platform for encouragement to get NGS sequencing funded in your country.”

Response 4. We further discussed this subject in the “Discussion” section:

“In addition to the high cost of this test, in many countries NGS sequencing is not covered by insurance, creating an important barrier to the disponibility of this technology. Consequently, only a portion of the population that can pay for the test will benefit from it.”

Reviewer 2 Report

This review article discuss the current understanding of biliary tract tumors (BTC), a rare form of liver cancer. The discovery of immune checkpoint inhibitors and a greater knowledge of tumor immunogenicity have developed in immunotherapy clinical studies for this circumstance, however not all patients benefit from this approach. This manuscript presents the most current findings about targeted and immunotherapy in the treatment of BTC, as well as the future prospects for this challenging disease.

Comments:

1-The authors of the review have shown the FGFR pathway (Figure1) and the differences between normal cells with IDH wild-type and cancer cells with IDH mutant (Figure2) as molecular pathways. These illustrations depict the downstream cascade of signaling pathways initiated by FGFR dimerization upon ligand binding and the aberrant activity caused by FGF binding. Even though these pathways may be well-known and widely discussed in the several scientific literature, the authors may have included these figures to support readers in comprehending the mechanics underpinning targeted therapy for biliary tract malignancies. Nevertheless, if the authors did not provide any new information to what is previously known about these paths, the figures may not offer much value to the review.

2- The authors must also discuss about immunotherapy resistance develop in response to treatment in patients. Downregulation or loss of expression of the programmed death-ligand 1 (PD-L1) protein, which is the target of immune checkpoint inhibitors such as pembrolizumab and nivolumab, is one cause of immunotherapy resistance. Without PD-L1 expression, the immune system cannot recognize and efficiently target cancer cells.

3-Immunotherapy in conjunction with chemotherapy as a possible therapeutic option for bile duct cancer should also be discussed in the text. The reason for this approach is that chemotherapy may increase the immunogenicity of tumors, therefore making them more recognized to the immune system and perhaps enhancing the effectiveness of immunotherapy.

4-The Authors need to also discuss about, epidermal growth factor receptor (EGFR) is a protein involved in cell division and proliferation. It is often overexpressed in a variety of cancers, including bile duct cancer, and is a possible therapeutic target. Monoclonal antibodies or tyrosine kinase inhibitors are examples of EGFR-targeted treatments (TKIs). 

5- To far, there have been very few reports showing bile duct cancer being treated with chimeric antigen receptor (CAR) T-cell therapy. CAR T-cell therapy is a kind of immunotherapy that includes genetically altering a patient's T cells to target and destroy cancer cells. Yet the text must cover some reports on the role of CAR-T treatment in BTC.

6- "Immunotherapy for Advanced Biliary Tract Cancer: Adding New Flavors to the Pizza" or any other title that emphasizes immunotherapy in BTC will be more suitable for this review.

Author Response

Comment 1. “1-The authors of the review have shown the FGFR pathway (Figure1) and the differences between normal cells with IDH wild-type and cancer cells with IDH mutant (Figure2) as molecular pathways. These illustrations depict the downstream cascade of signaling pathways initiated by FGFR dimerization upon ligand binding and the aberrant activity caused by FGF binding. Even though these pathways may be well-known and widely discussed in the several scientific literature, the authors may have included these figures to support readers in comprehending the mechanics underpinning targeted therapy for biliary tract malignancies. Nevertheless, if the authors did not provide any new information to what is previously known about these paths, the figures may not offer much value to the review.”

Response 1. We would also like to thank Reviewer #2 for his comments and suggestions. We discussed the decision to include Figures 1 and 2 to highlight the pathways related to the FGFR and IDH alterations. As these are uncommon mutations in other solid tumors, we think that is important to demonstrate to the readers how these pathways work. We also think that the figure is a better way to summarize this information without the need to also describe it in the text.

Comment 2. “2- The authors must also discuss about immunotherapy resistance develop in response to treatment in patients. Downregulation or loss of expression of the programmed death-ligand 1 (PD-L1) protein, which is the target of immune checkpoint inhibitors such as pembrolizumab and nivolumab, is one cause of immunotherapy resistance. Without PD-L1 expression, the immune system cannot recognize and efficiently target cancer cells.”

Response 2. We further discussed this subject in the “ICI Biomarkers” section:

“PD-L1 expression has been reported in 9.1 – 72.2% of patients with BTC, a membrane protein directly correlated with how immunotherapy works enhancing the immune response against the cancer cell85–87. Additionally, although the use of immunotherapy in the advanced scenario is approved regardless of PD1/PD-L1 expression, higher PD-L1 expression is associated with therapeutic response to immunotherapy, even in BTC73,88,89. However, even though there is a rationale to use PD-L1 as a biomarker of response, there is yet a long way to go. Alternately, there are data on lung cancer showing that low PD-L1 copy number tumors display reduced PD-L1 expression, reduced PD-L1 tumor cell staining, and an immunologic cold tumor microenvironment, thus could possibly leading to less response to immunotherapy.”

Comment 3. “3-Immunotherapy in conjunction with chemotherapy as a possible therapeutic option for bile duct cancer should also be discussed in the text. The reason for this approach is that chemotherapy may increase the immunogenicity of tumors, therefore making them more recognized to the immune system and perhaps enhancing the effectiveness of immunotherapy.”

Response 3. We further discussed this subject in the “Immune Checkpoint Inhibitors (ICI)” section:

“Also, the combination of immunotherapy with chemotherapy may represent a way to overcome the difficult scenario of advanced BTC treatment. Some studies have suggested that chemotherapy may upregulate checkpoint expression and change immune cell in-filtrate. Not only chemotherapy promotes tumor immunity by inducing immunogenic cell death but also acts by disrupting strategies that tumors use to evade immune recognition.”

Comment 4. “4-The Authors need to also discuss about, epidermal growth factor receptor (EGFR) is a protein involved in cell division and proliferation. It is often overexpressed in a variety of cancers, including bile duct cancer, and is a possible therapeutic target. Monoclonal antibodies or tyrosine kinase inhibitors are examples of EGFR-targeted treatments (TKIs).”

Response 4. We added the section “8. Epidermal growth factor receptor (EGFR)”:

“8. Epidermal growth factor receptor (EGFR)

EGFR is an essential transmembrane tyrosine kinase involved in activating the RAS/RAF/MAPK and AKT/mTOR signal transduction pathways, which are directly related to both cell proliferation and cell death91. Inhibition of EGFR has been a widely studied and ac-tionable target for several types of tumors, including non-small cell lung and colon cancer, using either tyrosine kinase inhibitors (TKIs) such as osimertinib and erlotinib, or antibodies such as cetuximab and panitumumab.

The expression of EGFR can vary depending on the analysis methodology. Shafizadeh et al. identified, through immunohistochemistry analysis, moderate or strong (+2/+3) EGFR ex-pression in approximately 59% of BTC samples, with a prevalence of 68%, 58%, and 33% in EHC, IHC and GB, respectively. When analyzed by FISH, 46% of the samples showed increased EGFR gene copy numbers. The same author identified worse 5-year survival outcomes among those expressing EGFR (20% vs. 60%).

Given the high prevalence and the prognostic impact of EGFR in BTC, several studies have evaluated the efficacy of targeting EGFR using either tyrosine kinase inhibitors (EGFR-TKIs) or monoclonal antibodies directed to EGFR (EGFR-mAbs). Although these tar-geted drugs have shown promising results in several types of tumors (NSCLC, colon, head, and neck), applying these treatments did not reproduce encouraging results in BTC. The BINGO trial, a randomized phase 2 trial, evaluated the association between cetuximab (EGFR-mAbs) with gemcitabine and oxaliplatin in patients with locally advanced (non-resectable) or metastatic BTC. The addition of the TKI did not result in significant gains in progression-free survival (6.1 vs. 5.5 months), overall survival (11 vs. 12.4 months), or objective response rate (24% vs. 23%) compared to the chemotherapy alone group. Furthermore, serious adverse events were reported in the antibody group. Similar results were found in the PICCA trial, which evaluated the gemcitabine and cisplatin plus panitumumab regimen when stratified by KRAS wild-type. When evaluating the association with the TKI erlotinib, the experimental group, although presenting a significant gain in objective response rate (30% vs. 16%, p 0.005), did not show a significant gain in progression-free survival (5.8 vs. 4.2 months, p 0.087) or overall survival (9.5 vs. 9.5 months, p 0.611).

Given the lack of statistically significant benefits in objective response rate, progression-free survival, and overall survival, a targeted therapy directed at EGFR (whether TKI or antibodies) has not been included in the therapeutic arsenal of BTC with overexpressed EGFR.”

Comment 5. “5- To far, there have been very few reports showing bile duct cancer being treated with chimeric antigen receptor (CAR) T-cell therapy. CAR T-cell therapy is a kind of immunotherapy that includes genetically altering a patient's T cells to target and destroy cancer cells. Yet the text must cover some reports on the role of CAR-T treatment in BTC.”

Response 3. We added the section “9. Chimeric antigen receptor T cell (CAR T Cell)”:

“9. Chimeric antigen receptor T cell (CAR T Cell)

Chimeric antigen receptors (CARs) are an innovative type of immunotherapy with a well-established role in treating hematologic malignancies and has been extensively studied in solid tumors. This treatment involves collecting T cells from the patient, which are then genetically engineered to express chimeric antigen receptors (CAR T-cells) capable of identifying specific proteins on the surface of tumor cells. Once infused into the patient, these modified cells can target and destroy the cancer cells.

Various receptors found on the surface of biliary tract cancer (BTC) cells have been targeted using CAR T-cell therapy. For instance, in a phase 1 trial, EGFR-specific CAR T-cell therapy was evaluated in 19 patients with unresectable or metastatic BTC refractory to chemotherapy. Fol-lowing infusion of a conditioning treatment with nab-paclitaxel and cyclophosphamide, 10 patients achieved stable disease and 1 complete response, leading to a 4-month progression-free survival. Among the primary side effects were fever, lymphopenia, and thrombocytopenia, attributed to the conditioning treatment.

In a phase 1 study by Feng et al., CART-HER2 immunotherapy was evaluated in 11 patients with BTC or advanced/metastatic pancreatic cancer with HER2 expression. Encouraging results were obtained, with a median progression-free survival of 4.8 months101. Other targets have also been explored using CAR T-cell therapy, such as Mesothelin, CD133, Claudin 18.2, and MUC-1, with promising results in patients with BTC and other solid tumors.”

Comment 6. “6- "Immunotherapy for Advanced Biliary Tract Cancer: Adding New Flavors to the Pizza" or any other title that emphasizes immunotherapy in BTC will be more suitable for this review.”

Response 6. We changed the title to “Immunotherapy and Targeted Therapy for Advanced Biliary Tract Cancer: Adding New Flavors to the Pizza.”

We would also like to highlight that we completed to modifications requested in the manuscript, like changing the abstract to a shorter one, adding the “citations”, “academic editor” and a Word version of Table 1. We also specified that the first and second authors contributed equally to the manuscript.

Round 2

Reviewer 2 Report

The authors have made all changes and responded to my suggestions.